

# Peptide presentation by HLA-DQ molecules is associated with the development of immune tolerance

Máté Manczinger[1,2,3] and Lajos Kemény[1,2]

[1] Department of Dermatology and Allergology, University of Szeged, Szeged, Hungary
[2] MTA-SZTE Dermatological Research Group, University of Szeged, Szeged, Hungary
[3] Synthetic and Systems Biology Unit, Institute of Biochemistry, Biological Research Centre, Hungarian Academy of Sciences, Szeged, Hungary

## ABSTRACT

HLA class II proteins are important elements of human adaptive immune recognition and are associated with numerous infectious and immune-mediated diseases. These highly variable molecules can be classified into DP, DQ and DR groups. It has been proposed that in contrast with DP and DR, epitope binding by DQ variants rather results in immune tolerance. However, the pieces of evidence are limited and controversial. We found that DQ molecules bind more human epitopes than DR. Pathogen-associated epitopes bound by DQ molecules are more similar to human proteins than the ones bound by DR. Accordingly, DQ molecules bind epitopes of significantly different pathogen species. Moreover, the binding of autoimmunity-associated epitopes by DQ confers protection from autoimmune diseases. Our results suggest a special role of HLA-DQ in immune homeostasis and help to better understand the association of HLA molecules with infectious and autoimmune diseases.

## INTRODUCTION

HLA molecules have an essential role in adaptive immune recognition (*Trowsdale, 2011*). HLA class II molecules reside on antigen presenting cells and bind protein fragments of endogenous and exogenous peptides (*Trowsdale, 2011*). The HLA-peptide complex then can be recognized by T cell receptors and induce enhanced immune response or tolerance (*Robinson & Delvig, 2002*). The equilibrium between immune-mediated elimination and tolerance is crucial for a healthy immune homeostasis (*Murphy et al., 2012*).

HLA class II molecules can be classified into DP, DQ and DR groups (*Trowsdale, 2011*). All molecules are made up by an alpha and a beta chain (*Jones et al., 2006*). Both chains of DP and DQ as well as the beta chain of DR are highly variable (*Murphy et al., 2012*). HLA class II molecules are associated with numerous diseases like autoimmunity, allergy and different kinds of infections (*Karnes et al., 2017*). To note, most diseases are associated with DQ and DR, while DP has a lower impact (*Karnes et al., 2017*). Consequently, our study focuses on the former two molecules.

Corresponding author
Máté Manczinger, manczinger.mate@med.u-szeged.hu

Previous studies reported important specialities in the localization, expression and function of DQ molecules. HLA-DQ is more abundant in the thymus than in the periphery (*Douek & Altmann, 2000*). Additionally, while DR is expressed in both the thymic cortex and medulla, DQ dominantly prevail in the cortex (*Ishikura, Ishikawa & Aizawa, 1987*) suggesting a special—rather suppressive—role of the molecule in immune homeostasis (*Altmann, Sansom & Marsh, 1991*). This expression pattern is the result of many transcriptional and post-transcriptional regulatory mechanisms (*Kobr et al., 1990*; *Miwa, Doyle & Strominger, 1987*). The suppressive behavior of DQ is suggested by several findings, some of which are related to the association of HLA-DQ alleles with autoimmunity. First, risk alleles for type 1 diabetes mellitus show a recessive behavior (*Todd, 1990*). It suggests that in case of heterozygosity, the other allele—being not associated with the disease—is able to induce tolerance in the thymus. Second, it has been shown that intrinsic stability of DQ molecules mediates susceptibility to autoimmune disorders (*Miyadera et al., 2015*). Unstable complexes cannot present self-peptides and, thus, central tolerance is not induced, which results in autoimmunity. Some associations between HLA-DQ and infectious diseases also suggest its dominant role in tolerance induction: it has been shown that HLA-DQ mediates non-responsiveness to *Schistosoma japonicum*, *Mycobacterium leprae* and BCG antigen through antigen-specific immune suppression (*Hirayama et al., 1987*; *Ottenhoff et al., 1990*; *Salgame, Convit & Bloom, 1991*). However, the predominantly suppressive role of DQ in immune homeostasis remained controversial as DQ-mediated proliferative responses are also reported for both autoimmune and infectious diseases (*Glanville et al., 2017*; *Van Lummel et al., 2014*).

Based on previous findings, we hypothesized that HLA-DQ has an essential role in the induction of tolerance mechanisms. We carried out systematic analyses of experimental and computationally predicted data to test four relevant predictions of the hypothesis: (i) DQ molecules bind more human epitopes, (ii) DQ-bound epitopes of pathogens are more similar to human proteins, than DR-bound ones, (iii) DQ and DR molecules recognize epitopes of different pathogen species and (iv) binding of autoantigens by HLA-DQ molecules confer protection from autoimmune diseases. Our findings highly supported the suppressive behavior of HLA-DQ molecules. Additionally, our results could help to better understand the adaptive immune recognition of pathogens and the development of autoimmune diseases.

## METHODS

### Determining epitope sequence similarity to human proteins

To determine epitope similarity to human proteins, *in vitro* HLA binding and T cell assay data were downloaded from the Immune Epitope Database (IEDB) (*Vita et al., 2015*). Positive assay results indicating the binding of 15 amino acids long epitopes by HLA-DQ and/or HLA-DR molecules were collected. Results for human epitopes were discarded. The human reference proteome was downloaded from Uniprot (*The UniProt Consortium, 2017*) and epitope sequences found in it were also discarded. Next, highly similar epitope sequences were excluded using an iterative method. First, a protein distance

matrix containing k-tuple distance values between all possible epitope pairs was generated with Clustal Omega (*Sievers et al., 2011*; *Yang & Zhang, 2008*). In each iteration, epitope pair (or pairs) with the smallest distance value were identified. For each member of the epitope pair (or pairs), the mean distance value to all other peptides was calculated and the epitope with the smallest value was excluded. The iterations were repeated until only larger than 0.5 k-tuple distance values remained in the matrix. This value corresponds to ~50% difference between the two sequences. This filtering process was carried out for HLA-DQ and HLA-DR epitope set separately resulting in 1476 and 4077 epitopes, respectively.

Each epitope sequence was decomposed to five amino acid long peptides (5-mers). For each 5-mer, the number of times it prevails in the human reference proteome was determined. For both DQ- and DR-associated epitopes, the proportion of 5-mers found for a given time in the human proteome were calculated. If one 5-mer could be detected more times in epitope sequences, all occurrences were taken into account. Similarly to a recent paper (*Trost et al., 2012*), we defined rare 5-mers in three different ways: 5-mers occurring 0 times, 5-mers occurring two or fewer times and 5-mers occurring five or less times in the human proteome. The epitope set containing significantly less rare 5-mers was considered to be more similar to human proteins. The level of significance was calculated with a randomization test. In each iteration, peptides of the original epitope set were randomly assigned to DR and DQ and the 5-mer analysis was carried out on these sequences. The epitope randomization and 5-mer analysis process was repeated for ten thousand times. *P* value was defined as the probability of having larger or equal difference between the proportion of rare or common 5-mers in DQ and DR-associated epitopes by chance than what we found (i.e., the number of such cases divided by the total number of iterations).

## Determining the species specificity of DQ and DR

HLA-II epitope sequences of pathogen species were downloaded from IEDB (*Vita et al., 2015*). Obligate intracellular pathogens were excluded from the analysis and highly similar sequences were discarded as described previously. Species with at least 25 epitopes available were selected for further analysis. Reference proteome of all species were downloaded from the Uniprot database (*The UniProt Consortium, 2017*). Epitopes found in only one proteome (i.e., species specific epitopes) were kept for further analysis. The previous filtering processes resulted in 1247 epitope sequences of 11 pathogens (Table S1). Common HLA-DRB1, DQA1 and DQB1 alleles were collected from the Common and Well Documented (CWD) Alleles Catalog (*Mack et al., 2013*). All HLA-DQA1-DQB1 allele combinations were generated and forbidden allele combinations were excluded (*Raymond et al., 2005*). The binding of each epitope by each common DRB1 allele or DQA1-DQB1 allele pair was predicted with the NetMHCIIpan-3.1 computer algorithm (*Andreatta et al., 2015*). We used the 10% rank percentile measure to define binding as suggested by a recent NetMHCIIpan server update. The fraction of epitopes bound by each allele was calculated for each pathogen. A matrix was created containing these values and hierarchical clustering was carried out to find similar alleles based on their recognition of species. For each pathogen, allele-specific recognition values were scaled and centered. Ward clustering

algorithm was used with the implementation of Ward's clustering criterion (*Murtagh & Legendre, 2014*). Clustering and visualization were carried out with pheatmap R library (*Kolde, 2012*).

To confirm in silico results with *in vitro* data, we used scoring matrices. Epitopes of each species were examined separately. Sequences longer than 15 amino acids were decomposed to 15 amino acid long sequences and score was calculated for each resultant peptide. Epitope sets created for the 5-mer analysis were used to generate scoring matrices. Before generating matrices, epitopes of the examined species were excluded from the epitope sets. The prevalence of each amino acid at each of the 15 positions of the epitope sequences was determined separately for the DQ and the DR-associated epitopes. This resulted in two different scoring matrices, which were used to assess the probability of binding the examined epitope by HLA-DQ and DR molecules. For each epitope of the examined species, we calculated two scores reflecting its binding by HLA-DQ and HLA-DR. The scores were determined by summing the values in the scoring matrices, which correspond to the given amino acids at the 15 positions of the examined epitope. For example, if an alanine was the first amino acid in the epitope sequence, we took the prevalence value for alanine at the first position from the scoring matrix. We repeated this process for all positions and summed the 15 values. Binding scores of the examined species' epitopes to HLA-DQ and HLA-DR were compared with Wilcoxon rank sum test. *P* values were adjusted using the Benjamini–Hochberg procedure (*Benjamini & Hochberg, 1995*).

## Determining the relationship between auto-epitope binding and susceptibility to autoimmune diseases

Associations between HLA alleles (or certain amino acids in allele sequences) and autoimmune diseases were collected from the PheWAS catalog (*Karnes et al., 2017*). These data were generated using HLA typing of a large population with detailed disease information. To our knowledge, PheWAS catalog is the only comprehensive source of HLA-disease associations. Associations with $P$ value less than $10^{-5}$ were considered to be significant as previously suggested (*Karnes et al., 2017*). Data about the following autoimmune diseases were collected: type 1 diabetes, Graves' disease, systemic lupus erythematosus, celiac disease, multiple sclerosis, primary biliary cirrhosis, systemic sclerosis, rheumatoid arthritis, juvenile rheumatoid arthritis, dermatomyositis and polymyalgia rheumatica. Only associations with exact disease terms were included in the analysis and terms that are only related to the diseases (for example ''Type 1 diabetes with ketoacidosis'') were excluded. Epitope sequences associated with each disease were collected from the IEDB (*Vita et al., 2015*). Sequences were discarded, if less than two references supported their role in disease development. After excluding diseases with lack of epitope sequence data, the following ones remained for further analysis: type 1 diabetes, Graves' disease, celiac disease, multiple sclerosis, primary biliary cirrhosis and rheumatoid arthritis. It is important to note that the catalog contains associations between diseases and individual DQA1 or DQB1 alleles, but not allele pairs. However, epitope binding of DQ molecules is determined by both alpha and beta chains (*Murphy et al., 2012*). As a solution, we selected all common allele pairs from the set we already generated (described previously),

which contain the given disease-associated allele. For each allele pair, we determined the fraction of disease-associated epitopes bound using the NetMHCIIpan algorithm as described previously (*Andreatta et al., 2015*). The median of these values defined the level of auto-epitope binding by the original disease-associated allele. Auto-epitope binding by DRB1 alleles were determined by calculating the fraction of disease-associated epitopes bound by the given disease-associated allele.

To examine associations between amino acids and autoimmune diseases, amino acid sequences of DQA1 and DQB1 alleles were downloaded from the IPD-IMGT/HLA database (*Robinson et al., 2015*). For each disease-associated amino acid, we selected common allele pairs containing the given amino acid in the given position. For each allele pair, we determined the bound fraction of auto-epitopes and calculated the median of these values to describe the level of auto-epitope binding associated with the given amino acid. The difference between auto-epitope binding by susceptibility, neutral and protective alleles was detected using Kruskal-Wallis test. Pairwise comparison was carried out with Wilcoxon rank sum tests. P values were adjusted using the Benjamini–Hochberg procedure (*Benjamini & Hochberg, 1995*).

## RESULTS

### HLA-DQ molecules bind more human epitopes than HLA-DR

The hypothesis that DQ molecules are associated with the induction of tolerance predicts that they bind more human epitopes than DR. To test this prediction, we collected results of all *in vitro* binding assays for human epitopes from the Immune Epitope Database (IEDB) (*Vita et al., 2015*). We selected 1079 epitope sequences, whose binding was tested to both DQ and DR alleles. We calculated the number of positive and negative assay results for DQ and DR and found a higher proportion of positive binding assays for DQ alleles (OR: 1.78, Fisher exact test P: $2*10^{-22}$). This result suggests a higher chance for the binding of human epitopes by DQ than by DR. To exclude the possibility that this is caused by a generally higher epitope binding capacity of DQ molecules, we collected assay results also for all non-human epitopes and selected 2289 sequences, whose binding was tested to both DQ and DR alleles. Reassuringly, we got the opposite result: the proportion of positive assays was higher for DR molecules than for DQ (OR: 1.36, Fisher exact test $P = 1.8*10^{-42}$).

### Epitopes of pathogens bound by HLA-DQ molecules are more similar to human proteins, than the ones bound by HLA-DR

We found a higher chance for binding human epitopes by DQ molecules than by DR. This suggests that epitopes of pathogens, which are bound by HLA-DQ might be more similar to human proteins than the ones bound by DR. To determine similarity of DQ and DR-associated epitopes to human proteins, we used an established method (*Trost et al., 2012*). Five amino acids long peptide segments (5-mers) are reported to be units of immunological recognition and protein-protein interactions (*Lucchese et al., 2007*). We downloaded positive *in vitro* MHC binding and T cell assay results for DQ and DR molecules from the IEDB (*Vita et al., 2015*). We selected 15 amino acid long sequences, excluded results for human epitopes and discarded highly similar epitope sequences (see

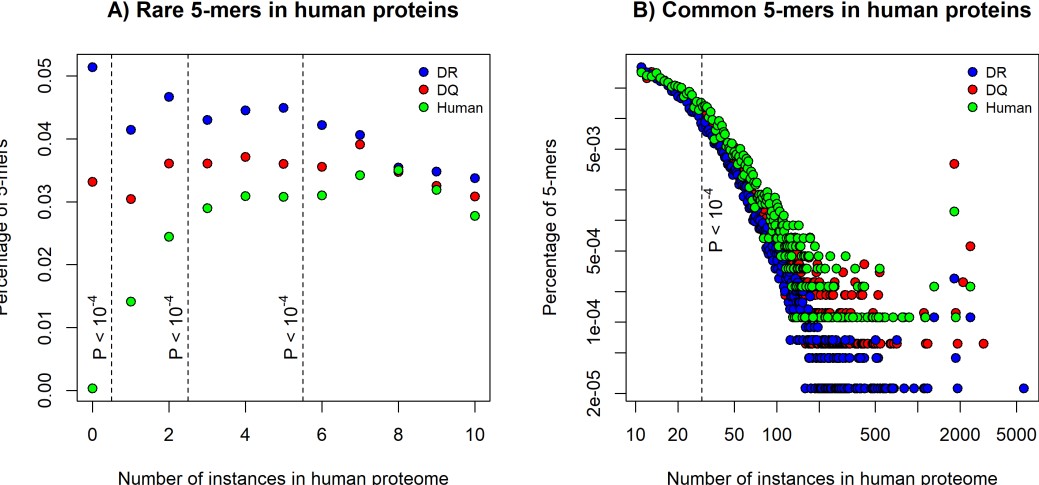

**Figure 1** **The percentage of rare and common 5-mers in DQ- and DR-associated epitopes.** The figures show the fraction of 5-mers found for certain times in the human proteome. DQ-associated epitopes contained (A) less rare 5-mers and (B) a higher number of common 5-mers than DR-associated sequences. 5-mer composition of 815 human epitopes (green) is also shown on the figures. Dashed lines represent different cutoffs used for defining (A) rare 5-mers and (B) common 5-mers. *P* values represent the probability of having the same or higher difference between the number of rare and common 5-mers in DQ and DR-associated epitopes by chance (see Methods). Note that in case of common alleles, both horizontal and vertical axes were log-transformed for better visualization.

Methods). The filtering process resulted in 1476 DQ and 4077 DR-associated epitopes. We decomposed each epitope to 5-mers and determined the number of times each 5-mer can be found in the human proteome. Then, for both DQ and DR-associated epitopes, we calculated the percentage of 5-mers that can be found for certain times in the human proteome. The percentage of rare 5-mers indicates the similarity of epitope set to the human proteome: the lower number of rare 5-mers can be found in epitope sequences, the more similar these epitopes are to human proteins. As expected, DQ-associated epitopes contained a significantly lower number of rare 5-mers, than DR-associated epitopes (Fig. 1A). Moreover, common 5-mers occurring 30 or more times in the human proteome could be found more frequently in DQ-associated epitopes (Fig. 1B).

### HLA-DQ and HLA-DR molecules bind different pathogen species

As DQ-associated epitopes show higher similarity to human proteins, DQ and DR might be responsible for the recognition of different pathogen species. To test this, we downloaded HLA-II epitopes of pathogen species from IEDB. We discarded peptide sequences of obligate intracellular pathogens as they are predominantly presented in an MHC-I dependent manner (*Hewitt, 2003*). We also excluded highly similar sequences from analysis (see Methods). We kept microbes having at least 25 documented epitopes in IEDB and predicted the binding of each epitope by 73 common HLA-DRB1 alleles and 168 common HLA-DQA1-DQB1 allele pairs using NetMHCIIpan-3.1. This is reported to be the most accurate prediction algorithm for MHC class II molecules (*Andreatta et al., 2017*). For each microbe, we determined the ratio of epitopes bound by each allele.
| DQ/DR | 0.16 | 0.22 | 0.34 | 0.78 | 0.8* | 1.26* | 1.81 | 2.06 | 2.12 | 2.75 | 3.26 |
|---|---|---|---|---|---|---|---|---|---|---|---|
| $P_{corr}$ | $1*10^{-30}$ | $2*10^{-30}$ | $4*10^{-26}$ | 0.02 | 0.002 | $8*10^{-5}$ | $3*10^{-5}$ | $3*10^{-22}$ | $2*10^{-6}$ | $1*10^{-8}$ | $1*10^{-25}$ |
| $P_{corr}$(Scoring matrices) | $2*10^{-9}$ | $3*10^{-7}$ | $4*10^{-5}$ | 0.07 | 0.01 | $2*10^{-42}$ | 0.91 | 0.56 | $1*10^{-34}$ | 0.18 | 0.05 |

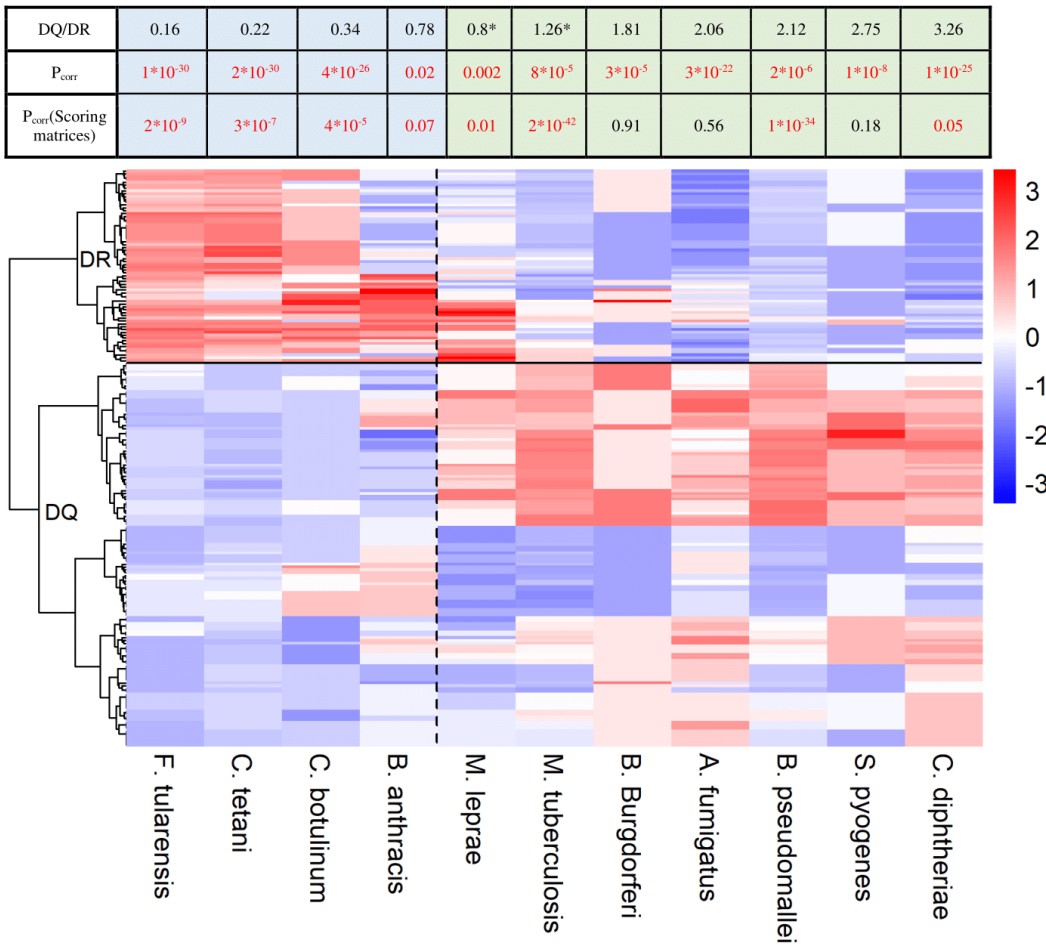

**Figure 2  HLA-DQ and DR molecules bind epitopes of different microbes.** The heatmap shows epitope binding by different DQA1-DQB1 allele pairs and DRB1 alleles color coded. In case of each species, colors represent the portion of epitopes recognized by each allele or allele pair. Each row corresponds to a DRB1 allele or DQ allele pair. Rows are clustered using hierarchical clustering (see Methods). DRB1 and DQ molecules are clearly separated based on their species preference (marked with a horizontal line). Epitopes of species on the left are preferred by DR (marked with blue color in the table) and on the right are preferred by DQ molecules (marked with green color in the table). Values were centered and scaled before being clustered and visualized. The ratio between the mean proportion of epitopes bound by DQ allele pairs and DR alleles (DQ/DR) is shown in the table for each species. Note that although computational prediction indicated similar recognition of *M. tuberculosis* (*) and *M. leprae* (*) by DQ and DR, analysis of in vitro data showed significantly higher binding scores of these species for DQ (Table S2). Consequently, they were classified into the DQ-associated group. $P_{corr}$ represents FDR-corrected *P* values of Wilcoxon rank sum test. $P_{corr}$ values lower than 0.1 were considered to be significant and highlighted with red color.

Next, we used hierarchical clustering to identify similarities between the species-preference of different alleles. DQ and DRB1 alleles clearly separated from each other making up two different clusters (Fig. 2). Additionally, DQ and DR alleles bound markedly different microbial species. We aimed to confirm these results with *in vitro* binding data. To this end, binding probability scores were calculated for each pathogen-associated epitope (see 'Methods' for details). Briefly, we determined the prevalence of each amino acid at the 15

amino acid positions for both DQ and DR-associated epitopes. We used epitope sets of the 5-mer analysis for this purpose. Amino acid prevalence values were then applied to calculate scores, which reflect the binding probability of a given epitope by DQ and DR molecules. Reassuringly, majority of results held when using empirical binding data (Fig. 2, Table S2).

### The binding of auto-epitopes by HLA-DQ molecules protects from autoimmune diseases

The preferred binding of human epitopes by DQ and previous evidence suggest, that these molecules might play an important role in the induction of tolerance to self-epitopes. Consequently, the binding of well-known epitopes of autoantigens (i.e., auto-epitopes) by DQ molecules might protect from, while lack of binding might confer susceptibility to autoimmune diseases. A straightforward prediction of these assumptions is that protective HLA-DQ alleles bind more, while risk HLA-DQ alleles bind less auto-epitopes than neutral alleles. To test these predictions, we collected 32 associations between 19 alleles of the DQA1, DQB1 and DRB1 loci and six autoimmune diseases from the PheWAS catalog (*Karnes et al., 2017*) (Table S3). For each disease, we collected disease-associated epitopes from IEDB (*Vita et al., 2015*). Epitope-binding by DQ is determined by both alpha and beta chains. Consequently, we calculated the characteristic level of auto-epitope binding by a given protective or risk allele by considering all DQA1-DQB1 combinations, in which the allele is included (see Methods). We found that the binding of disease-associated epitopes by DQ alleles negatively correlated with the allele-associated risk for autoimmune diseases (Fig. 3A). This relationship was independent of the autoimmune disease type and loci of DQ (DQA1 or DQB1) (Table 1). We got the same results, if we used the amino acid position dataset of PheWAS catalog instead of exact alleles (Fig. 3B, Table 1 and Table S3).

As expected, protective alleles bound a higher, while susceptibility alleles bound a lower portion of disease-associated epitopes than neutral ones (Kruskal-Wallis $P = 9 * 10^{-5}$, Fig. 4). No significant difference was found between autoantigen-binding by protective, neutral and susceptibility alleles of HLA-DR (Kruskal-Wallis $P = 0.18$).

## DISCUSSION

The equilibrium between immune defense mechanisms and tolerance is crucial for the homeostasis of immunity. HLA molecules have a fundamental role in the regulation of these processes as HLA-associated epitope presentation is one of the initial steps in the afferent arm of immune response (*Robinson & Delvig, 2002*). The different HLA class II loci are results of gene duplication events, and their evolution is relatively independent from each other (*Satta et al., 1994*; *Sommer, 2005*; *Valdes et al., 1999*). Several previous studies suggested a special—potentially suppressive—role of DQ molecules, but the evidence is controversial and limited (*Altmann, Sansom & Marsh, 1991*; *Hirayama et al., 1987*; *Miyadera et al., 2015*; *Ottenhoff et al., 1990*; *Salgame, Convit & Bloom, 1991*; *Todd, 1990*).

We carried out a large-scale analysis of epitope binding by DQ and DR molecules. We found a higher chance for binding human epitopes by HLA-DQ than HLA-DR. We
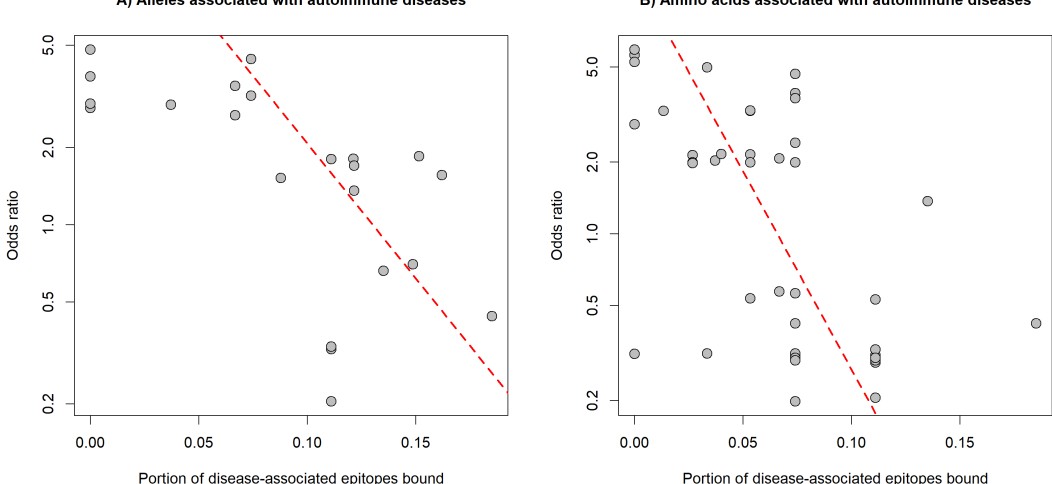

**Figure 3** **Binding of auto-epitopes by HLA-DQ is associated with protection from autoimmune diseases.** The binding of disease-specific auto-epitopes inversely correlated with the disease risk associated with (A) HLA-DQ alleles and (B) amino acids of HLA-DQ (Spearman's rho: $-0.69$ and $-0.64$, $P = 4 * 10^{-4}$ and $2 * 10^{-8}$, respectively). Horizontal axes indicate the portion of disease-associated auto-epitopes bound by (A) the given allele or (B) amino acid (see Methods). Vertical axes show the risk for autoimmune disease associated with (A) the given allele or (B) amino acid. Red dashed lines represent linear regression lines. Note, that vertical axes are on a logarithmic scale.

**Table 1** **The relationship between auto-epitope binding by HLA-DQ molecules and protection from autoimmune diseases is independent of disease types and DQ chains.** To test, whether the relationship between binding auto-epitopes and risk for autoimmune diseases is caused by disease- or DQ chain-specific differences in the epitope-binding of alleles, we constructed multivariate models: (1) $\log(\text{OR}) \sim Fr_{allele} + \text{Disease} + \text{Chain}$, (2) $\log(\text{OR}) \sim Fr_{amino\ acid} + \text{Disease} + \text{Chain}$, where $OR$ is the odds ratio for developing the given disease; $Fr_{allele}$ is the level of disease-associated auto-epitope binding by the DQ allele; $Fr_{amino\ acid}$ is the level of disease-associated auto-epitope binding by DQ alleles, which contain the given amino acid and; $Chain$ (i.e., DQA1 or DQB1) and $Disease$ are categorical variables. $Fr_{allele}$ and $Fr_{amino\ acid}$ showed negative effect on $OR$ after controlling for diseases and DQ chains. Significant relationship between predictor and response variables is marked with red color.

| | Allele associations | | | | Amino acid associations | | | |
|---|---|---|---|---|---|---|---|---|
| | Variable | Slope | Variance explained | P | Variable | Slope | Variance explained | P |
| | $Fr_{allele}$ | − | 0.4 | 0.01 | $Fr_{amino\ acid}$ | − | 0.39 | 0.008 |
| | Disease | NA | 0.17 | NA | Disease | NA | 0.08 | NA |
| | Chain | + (DQB1) | 0.02 | 0.47 | Chain | − (DQB1) | 0.08 | 0.004 |
| $R^2$ | 0.38 ($P = 0.046$) | | | | 0.5 ($P = 6*10^{-8}$) | | | |
| $N$ | 22 | | | | 61 | | | |
| BP test $P$ | 0.2 | | | | 0.11 | | | |

**Notes.**
Variance expained: The proportion of variance in $log(OR)$ explained by the given predictor variable; $P$: the probability of observing relationship between the predictor and response variables by chance; $R^2$: Total variance in $log(OR)$ explained by the model ($P$ corresponds to $F$-test $P$ value); $N$: number of associations between autoimmune diseases and alleles or amino acids; BP test $P$: the $P$ value for Breusch-Pagan test of heteroscedasticity, $P$ values larger than 0.05 suggest lack of heteroscedasticity.

utilized an established 5-mer based approach to compare pathogen-associated epitopes with human proteins and found that DQ-bound epitopes are more similar to self-proteins than DR-bound ones (Fig. 1). *Trost et al. (2012)*—using the same 5-mer based approach—found that bacteria causing chronic infections are more similar to the human proteome (i.e., they

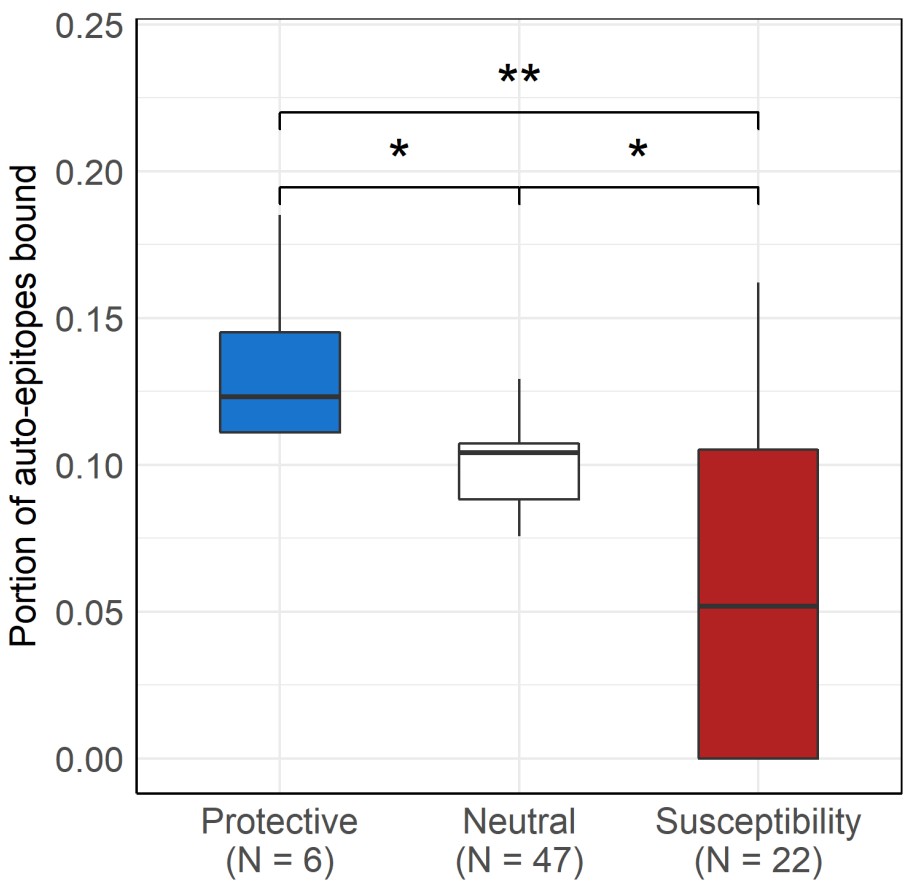

**Figure 4 Protective DQ alleles bind more, while risk DQ alleles bind less autoimmune disease-associated epitopes than neutral ones.** Boxplot indicates median (horizontal line), the first and third quartile (bottom and top of boxes) and minimum and maximum values (vertical lines). The bound portion of auto-epitopes of the given disease that is associated with the given allele is shown for protective and susceptibility groups. The portion of all auto-epitopes bound by each allele is shown for the neutral allele group. * $P_{corr} < 0.005$ (pairwise Wilcoxon test), ** $P_{corr} < 0.01$ (pairwise Wilcoxon test)

contain a lower number of rare 5-mers) (*Trost et al., 2012*). This is in line with another study suggesting that rare 5-mers cause a more intense immune response than common ones (*Patel et al., 2012*). We characterized the binding of pathogen-associated epitopes by DQ and DR and found that DQ and DR molecules present peptides of rather different microbial species (Fig. 2). While DR molecules bind pathogens associated with acute infections, DQ alleles also recognize epitopes of pathogens causing chronic infections (*M. tuberculosis, M. leprae*) (*Russell, 2011*; *Yamamura et al., 1991*) or evading immune system and having a high relapse rate after therapy (*B. pseudomallei*) (*Currie et al., 2000*).

Finally, our results suggest that DQ molecules have an essential role in the development of central tolerance by presenting self-epitopes. This is in line with previous findings: (i) non-stable binding of peptides by HLA-DQ resulted in thymic escape of autoreactive T-cells (*Dendrou et al., 2018*) and (ii) autoimmune dermatitis developed in transgenic mice expressing an HLA-DQ ortholog I-A$^b$ complex, which can bind only one epitope

(*Logunova et al., 2005*). In these mice, autoreactive cells could be maintained due to the lack of negative selection of T cells, which recognize self-peptides by MHC-II dependent presentation. We examined six different autoimmune diseases and found that the hazard ratio associated with a given HLA-DQ allele inversely correlated with the proportion of bound disease-associated epitopes (Fig. 3, Table 1). Additionally, neutral DQ alleles bound less epitopes than protective and more epitopes than risk alleles (Fig. 4). This result indicates, that a certain level of auto-epitope binding by DQ molecules is needed for a healthy immune-homeostasis. Alleles binding less auto-epitopes might allow thymic escape of self-reactive T cells and make the individual susceptible to autoimmune diseases.

## CONCLUSIONS

Previous pieces of evidence suggested a suppressive role of HLA-DQ molecules in immune homeostasis, but this hypothesis remained controversial. We tested relevant predictions of the hypothesis. We found that DQ molecules bind more human epitopes than DR. Accordingly, pathogen-assocated epitopes bound by DQ are more similar to human proteins than the ones bound by DR. DQ molecules bind mainly epitopes of pathogens associated with chronic or relapsing infectious diseases. This indicates the importance of DQ-mediated tolerance induction for the immune evasion of pathogens. Our results also suggest that DQ molecules might have a more important role in inducing tolerance than in activating proliferative and destructive responses during the development of autoimmune diseases. All of our findings suggest an essential role of HLA-DQ molecules in tolerance formation and might help to better understand the role of HLA molecules in the development of infectious and autoimmune diseases.

### Funding
Máté Manczinger was supported by the UNKP-17-4 New National Excellence Program of the Ministry of Human Capacities. The funders had no role in study design, data collection and analysis, decision to publish, or preparation of the manuscript.

### Grant Disclosures
The following grant information was disclosed by the authors:
New National Excellence Program of the Ministry of Human Capacities: UNKP-17-4.

### Competing Interests
The authors declare there are no competing interests.

### Author Contributions
- Máté Manczinger conceived and designed the experiments, performed the experiments, analyzed the data, contributed reagents/materials/analysis tools, prepared figures and/or tables, authored or reviewed drafts of the paper, approved the final draft.
- Lajos Kemény conceived and designed the experiments, authored or reviewed drafts of the paper, approved the final draft.

## Data Availability

The raw data are provided in Tables S1 and S2.

## Supplemental Information

Supplemental information for this article can be found online at http://dx.doi.org/10.7717/peerj.5118#supplemental-information.

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
