# Peer review of "Peptide presentation by HLA-DQ molecules is associated with the development of immune tolerance"

_PeerJ, doi:10.7717/peerj.5118_

## Round 0.1 · original submission · Major Revisions

Experimental confirmation data is required to appreciate the bioinformatic analysis presented in this manuscript.

Reviewer 1 ·

Basic reporting

As a whole the manuscript entitled “Peptide presentation by HLA-DQ molecules is associated with the development of immune tolerance” is well written in a logical way. Cited references in the Introduction are sufficient to provide the readers with the necessary information and problems about the role of the different HLA molecules. There are 3 figures, 1 table and supplementary materials from 2 tables. The proposed hypotheses are not novel and have only confirmatory character.

Experimental design

This article reports results from the bioinformatic modelling of the binding ability of immune epitopes (taken from database) to the HLA-DR and HLA-DQ molecules. The used methods for the performed analyses are well described. Although the results support previous findings about the role of HLA-DQ for tolerance induction, the lack of any experimental data reduces the merit of the manuscript.

Validity of the findings

Therefore, the conclusions in this case could be treated as speculative, but not as confirmatory evidences. Performed analyses by using 5-mer epitopes and 50% binding ability are far away from the real situation.
The problem is that neither in the title nor in the abstract is written that this is only modelling, which could mislead the readers that the results are experimental. In addition, the results generated by these analyses are not novel. In best case only confirm some of the experimental data published from other authors such as epitope binding and association of HLA-DQ with certain autoimmune diseases. I think that the theoretical models have to precede the experimental results, not opposite (to confirm them).

It is not clear what means the sentence:
“… epitopes presented by DQ molecules are more similar to human proteins…”. which is mentioned few times. Some of these epitopes are part of human proteins. How the epitopes are distinguished whether they are human or not. What about the epitopes that have the same sequence in several organisms?

Additional comments

May be the manuscript will be improved if in the title and in the abstract is stated that this is only modeling (bionformatic analyses) and the proposed conclusion are based on these analyses (speculative), which need to be confirmed experimentally.
Alternatively, these bioinformatic analyses have to be supported at least partially with experimental data. For example with synthetic peptides (epitopes) in vitro/ex vivo or in vivo with animal models.

Reviewer 2 ·

Basic reporting

The manuscript meets all criteria of basic reporting.

Experimental design

no comments

Validity of the findings

no comments

Additional comments

In this paper Manczinger and Kemeny present a large-scale comparative analysis of epitopes presented by DQ and DR variants of HLA class II proteins. They found that epitopes presented by DQ molecules are more similar to human proteins as compared to those presented by DR molecules and showed around 10% overlap, suggesting lack of functional redundancy. In addition, DQ and DR molecules present peptides derived from different microbial species. The data presented in this manuscript may contribute to the understanding of MHC II biology, but I have some concerns regarding this study.
The authors claim that “ the binding of autoimmunity-associated epitopes by DQ confers protection against autoimmune diseases”. I think, this is not entirely correct.
Different alleles of DQ as well as DR have different impact on development of autoimmune diseases. For instance, certain of HLA-DQ heterodimers have central role in celiac disease and type 1 diabetes pathogenesis. Ninety percent of Caucasian patients with celiac disease express HLA-DQ2.5cis encoded by HLADQA1*0501-DQB1*0201 or DQ2.5trans onHLA-DQA1*0505 DQB1*0301/DQA1*0201-DQB1*0202 haplotypes. In contrast, approximately 20–30% of the healthy Caucasian population is HLA-DQ2 positive.
Therefore, it would be more useful to compare epitope-binding of DR and DQ alleles associated with risk to autoimmunity vs protection against autoimmunity (Fig.1).
From the microbe-derived epitope binding study (Fig. 2) the authors excluded peptide sequences of commensals. Since, it has been postulated that HLA-DQ may also have impact on the intestinal microbiome and, consequently, to development of autoimmunity, I think, it is reasonable to include these epitopes as well.

---

## Round 0.2 · accepted · Accept

Authors have satisfactorily addressed the concerns raised during the review process of their manuscript.

# Reviewer 1 ·

Basic reporting

The revised manuscript meets the PeerJ criteria.

Experimental design

Now, in addition to the predictive bioinformatic analyses are integrated also data from in vitro experiments.

Validity of the findings

The manuscript is improved significantly.

Additional comments

No comments